biotechnology/environmental science/ plant science

membrane vesicles, foliar fertilization, nanofertilization, agriculture

**Author for correspondence:**
M. Carvajal
e-mail: mcarvaja@cebas.csic.es

This article has been edited by the Royal Society of Chemistry, including the commissioning, peer review process and editorial aspects up to the point of acceptance.

# Nanobiofertilization as a novel technology for highly efficient foliar application of Fe and B in almond trees

J. J. Rios, L. Yepes-Molina, A. Martinez-Alonso and M. Carvajal

Group of Aquaporins, Plant Nutrition Department, Centro de Edafologia y Biologia Aplicada del Segura, CEBAS-CSIC, Campus Universitario de Espinardo, E-30100, Murcia, Spain

MC, 0000-0001-7321-4956

Nanofertilization is postulated as a new technology to deal with the environmental problems caused by the intensive use of traditional fertilizers. One of the aims of this new technology is to improve foliar fertilization, which has many environmental advantages, but currently there are numerous factors that limit its efficiency. In this research, the objective was to study the potential of membrane vesicles derived from plant material as nanofertilizers of iron (Fe) and boron (B) for foliar application in almond trees (*Prunus dulcis* L.). The results show that the application of vesicles caused invaginations in the plasma membrane of the leaf cells. Also, the increase in leaf B and Fe was greater when these elements were applied in an encapsulated form rather than in a non-encapsulated form. The distribution of these elements in leaf tissues indicated the existence of an intracellular element transport pathway and accumulation areas, enabling greater element entry and mobility.

## 1. Introduction

For many years, there has been a need to optimize the use of chemical fertilizers in farms, an extensive practice that provides nutrients to crops to boost growth and yields [1]. The intensive use of soil fertilizers can cause important environmental problems in natural ecosystems, such as water pollution and soil degradation [2]. This points out the need to design new techniques and products for the development of more efficient fertilization protocols [3].

Nanofertilization could be one of the best options for the development of efficient fertilization, since it is highly specific, involves the application of smaller quantities of product and thus avoids pollution problems [4]. In this sense, the most studied nanofertilizers are nanoparticles. These can be made by different

physical–chemical approaches: (i) through the physical breakage of larger particles, or (ii) by joining together smaller particles of elements or compounds to achieve a certain size [5]. Also, nanocarriers are considered nanofertilizers since they are made from materials—organic material such as lipid, polymers or carbon nanotubes; or inorganic material such as different oxides—that bind mineral nutrients. Thus, the nanocarriers carry the elements and they are classified, according to the nutrients that they contain, as micronutrient or macronutrient nanofertilizers [4]. Furthermore, there is another type of nanofertilizer, the so-called nanobiofertilizers, that features the integration of biofertilizers, prepared with microorganism or plant extracts, into nanoparticles or nanostructures [6].

Currently, one of the most important fields in the improvement of agricultural fertilization is the enhancement of the efficiency of foliar fertilization, mainly with micronutrients. Foliar fertilization is an agricultural practice that is taking on great importance, since theoretically it is environmentally friendly, due to the fact that the nutrients are applied directly to plant tissues and in small amounts [7]. But, foliar fertilization has many factors that delimit its efficiency, such as the relative humidity and temperature [8]. Obviously, leaves are not designed for the absorption of nutrients from external areas. Hydrophilic compounds have more difficulty in passing into leaves than hydrophobic ones, but both could enter, mainly though stomata [9–11]. Thus, nanotechnologies will be able to have a fundamental role in increasing the efficiency of fertilization, mainly in foliar application. The size and charge of particles are important characteristics influencing their passage through the outer layer of the leaf. In fact, the small size of nanofertilizers increases the area of contact with the environment and with the leaf surface; in consequence, entry through the stomatal pore should be high, since its width is around 1 μm [12].

Micronutrients are elements that are essential to the plant but are required at low concentrations. Boron (B) and iron (Fe) are two important micronutrients in plants. Boron is involved in plant cell structure, biosynthesis and lignification, and is also crucial to the maintenance of vascular vessels functionality [13]. Also, Fe is involved in many fundamental processes—such as respiration (as it is required in membrane transport of electrons), photosynthesis and DNA synthesis. Also, it has structural functions, in Fe-S clusters and Fe-binding sites [14–16]. Boron and Fe deficiencies are among the most widespread deficiencies of plant micronutrients in agriculture [17,18]. Both elements have poor movement within plants and, once assimilated within the mature leaves, they are hardly remobilized towards the young or reproductive zones [11,19]. So, deficiency symptoms are observed in the young parts of plants and the common solution is soil application of Fe (as Fe chelates) and B (as borax) [18,20], but repeated treatments can cause soil contamination problems.

A new type of nanobiofertilization could be derived from the capacity to obtain hydrophobic structures from biological materials, using their natural ability to bind ions of micronutrients [21]. Therefore, the coat would be fully biodegradable, avoiding an impact on the environment. However, the investigation in this matter is very limited. Among the few studies reported, the use of polymer hydrogels as a network structure, to delay the mineral release and decrease evaporation losses, has been described [22,23].

*Prunus* L. spp. are the most important fruit trees in agriculture. Indeed, almond trees (*Prunus dulcis* L.) have a strong socioeconomic role in the Mediterranean countries. They have traditionally been cultivated as a rain-fed crop and their yield is usually low [24]. However, in recent years, as the price of almonds has increased due to their scarcity, a new interest in this crop has developed among farmers. However, almonds are cultivated in depleted soils with low fertility and the fertilization of the crops must be as efficient and economic as possible, along with great respect for the environment, avoiding the excessive application of nutrients [25]. Furthermore, the cultivation conditions (lack of rain) render the application of fertilizers to the soil ineffective, making foliar application the most advisable option to avoid or correct nutritional deficiencies in these trees [26].

Finally, a previous study by our group demonstrated the novel use of proteoliposomes, with high encapsulation efficiency and high delivery of Zn into protoplasts [27]. In this current work, we investigated the possibility of using the proteoliposomes as Fe and B nanobiofertilizers in almond. For this, the characteristics of the carriers containing Fe and B were studied. Also, their effectivity as a fertilizer was determined by comparison with free elements fertilization, studying the microlocalization and movement of the elements throughout the plants after foliar application.

# 2. Material and methods

## 2.1. Experiment under controlled conditions

Clones of the *Prunus dulcis* L. variety Avijor were acquired from Jodar Nurseries S.L. in Murcia (Spain). The plants were grown for 5 days in a substrate composed of peat : coconut-fibre : perlite (5 : 4 : 1). They were

then transferred to hydroponic culture in 12 l boxes (5 plants in each of 8 boxes, = 40 plants) filled with Hoagland solution, pH 5.5. The solution was continuously aerated and was changed every week. The composition of the solution was 6 $KNO_3$, 4 $Ca(NO_3)_2$, 1 $KH_2PO_4$ and 1 $MgSO_4$ (mM), and 25 $H_3BO_3$, 2 $MnSO_4$, 2 $ZnSO_4$, 0.5 $CuSO_4$, 0.5 $(NH_4)_6Mo_7O_{24}$ and 20 Fe-EDDHA (µM). The plants were grown in a culture chamber with controlled conditions: a cycle of 16 h of light and 8 h of darkness, with a temperature of 25°C and 20°C and relative humidity of 70% and 80%, respectively. The photosynthetically active radiation (PAR) was 400 µmol $m^{-2}$ $s^{-1}$, provided by fluorescent tubes (Philips TLD 36 W/83, Jena, Germany and Sylvania F36 W/GRO, Manchester, NH, USA) and metal halide lamps (Osram HQI, T 400 W, Berlin, Germany). After 10 days, deficiency treatments were applied: two boxes continued with full nutrient solution, three boxes had Fe deficiency (0 µM Fe added) and three boxes had B deficiency (0 µM B added). The plants were grown under these conditions until deficiency symptoms appeared progressively in the young leaves (approximately 20 days).

Once the deficiency symptoms had appeared, foliar applications were carried out for leaves that had expanded under B or Fe deficiency, avoiding the meristematic area. The treatments were applied approximately 2 h after the onset of the light period in the growth chamber. The experiment had a completely randomized design and was repeated three times. The solutions applied were (i) 0.02% Fe (from $FeSO_4$), nanoencapsulated, (ii) 0.02% free Fe (from $FeSO_4$), (iii) 0.04% B (from $H_3BO_3$), nanoencapsulated, and (iv) 0.04% free B (from $H_3BO_3$). In the control treatment, water was applied. All the treatments included 0.1% of a non-ionic organo-silicon surfactant [27]. The treatments (approximately 100 µl per leaf) were applied to the abaxial surface of the leaves using a sprayer, in an amount sufficient to completely wet the surface while avoiding dripping. Leaf samples were taken for mineral and microscopic analysis 60 min and 24 h after the application. The leaves were washed with distilled water containing 1% non-ionic soap to remove the fertilizer adhering to the surface.

## 2.2. Experiment under field conditions

Trees (15) from three different experimental fields—between 400 and 534 m above sea level and subjected to a Mediterranean climate—were selected for this experiment. The foliar treatments applied were: (i) 0.02% Fe (from $FeSO_4$), nanoencapsulated, (ii) 0.02% free Fe (from $FeSO_4$), (iii) 0.04% B (from $H_3BO_3$), nanoencapsulated, and (iv) 0.04% free B (from $H_3BO_3$). In the control treatment, water was applied. All the treatments, including the control, were applied with 0.1% of surfactant [27]. To randomize the treatments, five branches of each tree were selected around all the cardinal points. Each of the five treatments was applied randomly to each of the selected branches of each tree (three trees per treatment). The orientation of each applied treatment was varied in each tree, to avoid the cardinal effect. These applications were performed in the early morning, by spraying the fertilizers onto leaves until they were completely wet. Leaf samples were taken for mineral analyses 24 h after the application.

## 2.3. Vesicles isolation

Vesicles were extracted according to [27]. Leaves (100 g) were cut into small pieces before vacuum filtering with 0.5 g of PVP and 160 ml of buffer containing 0.5 M sucrose, 1 mM DTT, 50 mM HEPES and 1.37 mM ascorbic acid, at pH 7.5. Then, the sample was homogenized using a blender and filtered through a nylon mesh with a pore diameter of 100 µm. The filtrate was centrifuged at $10\,000 \times g$ for 30 min, at 4°C. The supernatant was recovered and centrifuged for 35 min at $100\,000 \times g$, at 4°C. Afterwards, the pellet obtained was suspended in 500 µl of 5 mM phosphate buffer and 0.25 M sucrose at pH 6.5. The protein concentration in the vesicles fraction was determined with an RC DC Protein Assay kit (BioRad, California, USA), using BSA as the standard.

## 2.4. Encapsulation

The Fe (as $FeSO_4$) and B (as $H_3BO_3$) were dissolved at concentrations of 2% and 4%, respectively, in distilled water. These solutions were vortexed for 30 s with the vesicles fraction (1 : 1 v : v) and were reconstituted for 10 min.

The percentage of each element encapsulated in the vesicles was determined by the direct measurement of the Fe or B concentration in the original solution and in the supernatant after centrifugation of the vesicles of the nanofertilizer (figure 1).

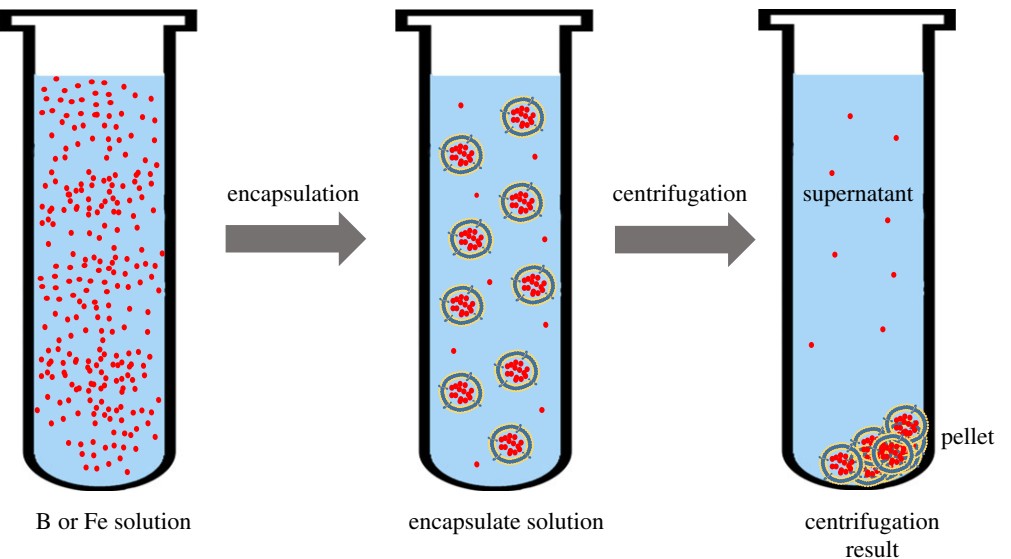

**Figure 1.** The encapsulation process.

## 2.5. Size and charge of vesicles

The average size (nm), polydispersity index (0–1) and Z potential (mV) of the vesicles containing Fe or B were checked using dynamic light scattering (DLS), through intensity measurements with a Malvern ZetaSizer Nano XL machine (Malvern Instruments Ltd, Orsay, France). This allowed the analysis of particles with a size range from 1 nm to 3 μm.

## 2.6. Osmotic water permeability

The osmotic water permeability (Pf) was measured as the velocity of the volume adjustment of the plasma membrane vesicles after the osmotic potential of the surrounding medium was changed. The volume of the vesicles was followed by 90° light scattering at $\lambda_{ex} = 515$ nm. The samples used were vesicles with encapsulated Fe or B. The measurements were carried out at 20°C in a PiStar-180 spectrometer (Applied Photophysics, Leatherhead, UK), as described previously by [28].

## 2.7. Transmission electron microscopy

Sections ($2 \times 2$ mm) were taken from different treated and control (water and surfactant) almond leaves and were fixed chemically with MacDowell fixative at 4°C for 1 h under vacuum conditions and then for 11 h in a refrigerator. After this, they were left for 6 h in a washing solution, which was changed every 2 h. The sections were dehydrated with ethanol and epoxypropane at 22°C and then embedded in Epon. The blocks were sectioned on a Leica EM UC6 ultramicrotome. The sections were collected on Formvar-coated copper grids and stained with uranyl acetate followed by lead citrate. The sections were examined using a JEOL 1011 transmission electron microscope and photographed with a GATAN ORIUS SC200 digital camera. For each sample, 8–10 ultra-thin sections were examined, in each of two independent experiments.

## 2.8. Elements distribution images mapping

Foliar-treated and deficient leaves were washed twice with distilled water and then blotted dry with filter paper. Leaf pieces ($1 \times 5$–6 cm) were embedded in 4% agar and transversal leaf cross-sections were obtained by hand-cutting. The cross-sections were quickly freeze-dried. To maintain the sections in good condition, they were dried between slides.

The sections were transferred to a stub. Sections of Fe-treated or deficient leaves were surrounded by Cu, to improve the load and enhance the measurement of the Fe. Since B is a light element, the B sections were coated with 5 mm of platinum, under vacuum, to achieve optimal resolution.

**Table 1.** Vesicles parameters after element encapsulation.

| element encapsulated | % encapsulation | vesicles size (nm) | polydispersity index | Z potential (mV) | Pf ($\mu$m s$^{-1}$) |
|---|---|---|---|---|---|
| Fe | 76 | $320 \pm 40.2$ | $0.34 \pm 0.04$ | -20.5 $\pm$ 0.91 | $21.3 \pm 0.53$ |
| B | 80 | $301.15 \pm 32.5$ | $0.28 \pm 0.02$ | -21.8 $\pm$ 0.72 | $20.1 \pm 0.73$ |

For mapping of the elements, the sections were observed by field emission scanning electron microscopy (SEFEM) coupled with energy-dispersive X-ray spectroscopy (EDX). The images were taken at 20 kV and a magnification of ×400–500.

## 2.9. Ion concentrations

For ion analysis, finely ground samples of lyophilized leaves were digested in a microwave oven (CEM Mars Xpress, North Carolina, USA), by $HNO_3 : HClO_4$ (2 : 1) digestion. The elements were detected by inductively coupled plasma (ICP) analysis (Optima 3000, PerkinElmer).

The concentrations of ions in the vesicles were determined after centrifugation of the vesicle preparations. The pellets obtained were dried and digested and the ions were determined by ICP, as described previously.

## 2.10. Statistical analysis

The statistical analysis of the ion concentrations measured in the leaves of plants grown under controlled conditions was carried out with 105 values (5 plants × 7 treatments × 3 analytical replicates). For the field conditions experiment, the analysis involved 180 values (15 plants × 4 treatments × 3 analytical replicates). All values were analysed by one-way analysis of variance (ANOVA), at the 95% confidence level, using the software SPSS Release 18 for Windows (SPSS Inc., Chicago, IL, USA). The statistical significance was considered as $^*p < 0.05$; $^{**}p < 0.01$; $^{***}p < 0.001$; and n.s.—not significant. Also, Duncan's test at $p \leq 0.05$ was chosen to determine the significance of differences between treatments. The values presented are means ± s.e.

# 3. Results

## 3.1. Vesicles parameters after encapsulation of elements

The characterization of the vesicles is shown in table 1. The percentage encapsulation observed was high for both elements: 76% for Fe and 80% for B. The measurements of vesicle size revealed that the population was homogeneous, with average hydrodynamic diameters of 320.0 nm (Fe vesicles) and 301.1 nm (B vesicles) and low polydispersity indexes, 0.34 for Fe vesicles and 0.28 for B vesicles. Hence, there was no significant difference in size between the two types of the vesicle (encapsulated with Fe or B). The Z potential was also determined for vesicles containing Fe or B, in suspension in 5 mM phosphate buffer and 0.25 M sucrose, at pH 6.5. There was a negative electric charge on the surface of the vesicles, −20.5 mV for Fe vesicles and −21.8 mV for B vesicles. Also, the osmotic water permeability (Pf) of the vesicles containing encapsulated Fe or B was similar: 21.3 $\mu$m s$^{-1}$ and 20.1 $\mu$m s$^{-1}$, respectively (table 1).

## 3.2. Effects of vesicles application in tissues and cells

Figure 2 shows leaf sections as observed by transmission electron microscopy (TEM). As can be seen, the cell structure of untreated (control) almond leaves was not altered when free Fe or B was applied (figure 2a,c). However, the encapsulated elements produced many areas with several invaginations of the plasma membrane of the abaxial epidermis and spongy mesophyll cells of the leaves after foliar application (figure 2b,d). The black arrows show the invaginations in these sections. No other alterations were observed.

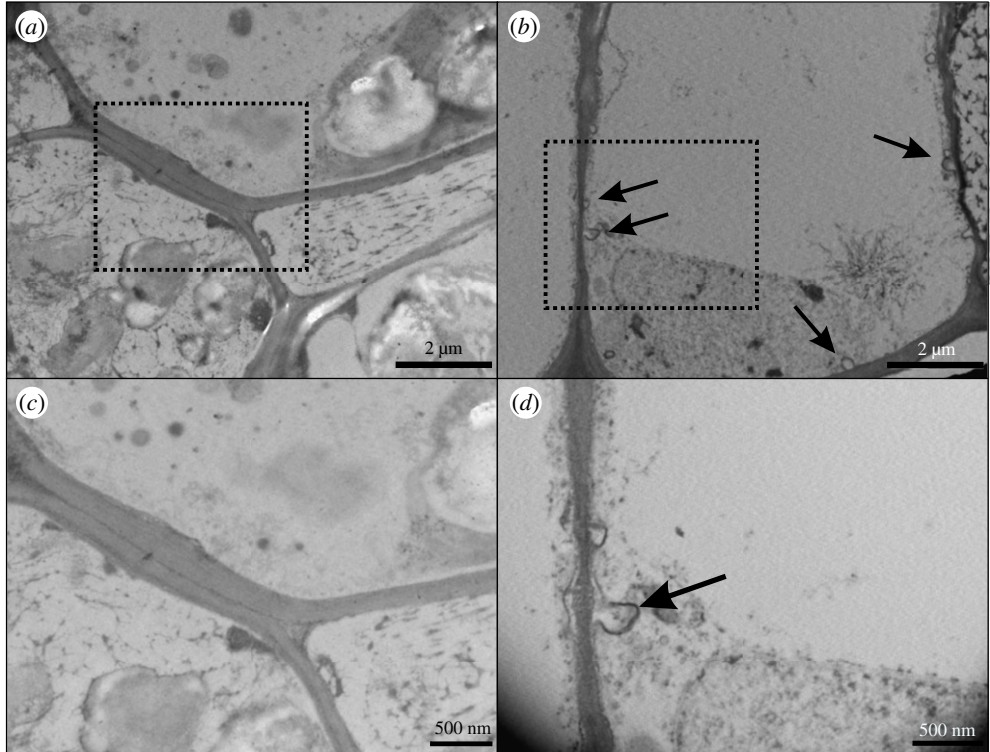

**Figure 2.** TEM of *Prunus dulcis* L. leaves. (*a*) Untreated leaves. (*b*) 1 h after nanofertilizer application. (*c*) Square from A, with greater magnification, showing plasma membrane (PM) details of control plant. (*d*) Square from B, with greater magnification, showing PM details of fertilized plant. Black arrows show invaginations of PM.

## 3.3. Foliar application

The concentrations of Fe and B in almond leaves after application under controlled conditions are shown in table 2. Significant differences were produced by the application of both elements. With respect to Fe, the values were low in deficient leaves, but were enhanced by treatment with free Fe and, especially, the Fe nanobiofertilizer: the increase in nanobiofertilized plants was 20% relative to those treated with free Fe. With respect to B, the results reveal the same pattern as for Fe, although the nanobiofertilization with B gave a much higher leaf concentration (87.31%) than the free form.

Also, we determined the Fe and B concentrations in the apex, which was not treated directly. The results show a statistically significant increase in concentration in this area after Fe or B application, with respect to deficient plants (table 2). The application of nanobiofertilizer gave a higher concentration of both elements, Fe and B, than when they were applied in free forms. Also, the % movement of Fe and B (calculated from data of tables 2 and 3) was higher with nanobiofertilizer than when the free elements were applied (17.5% of free Fe versus 21% of Fe-nanobiofertilizer and 20% of free B versus 36% of B-nanobiofertilizer).

The results of the experiment conducted under field conditions show the same pattern as in control conditions. We observed significant increases in the concentrations of Fe and B after their foliar application, relative to control leaves. The Fe- and B-nanobiofertilizers gave significantly higher concentrations than their respective free forms, being almost double in the case of B (table 3).

## 3.4. Elements mapping

The Fe and B distributions in transversal sections of almond leaves, after their foliar application, are shown in figures 3 and 4. The elements are represented in different colours, red for Fe and blue for B. Thereby, the Fe mapping in the sections shows that foliar application of free Fe yielded a central distribution, around the upper spongy mesophyll tissue and close to the vessels, with an increase in the area and intensity of the red colour relative to the control leaves (figure 3*b,e*). In the case of nanoencapsulated Fe, the red colour was much brighter and more extensive, being found throughout the lower part of the leaf, from the central vessel to the spongy mesophyll and the abaxial epidermis,

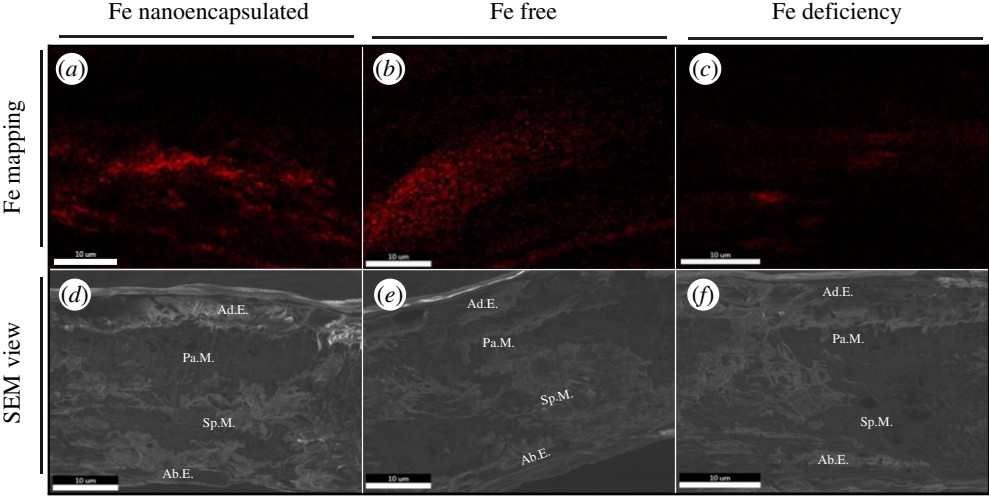

**Figure 3.** Fe mapping of cross-sections of *Prunus dulcis* L. leaves by FESEM-X-EDS. The red colour represents the Fe signal. (*a*) 24 h after nanofertilizer application. (*b*) 24 h after free element application. (*c*) Deficient. SEM image of almond leaf section (*d,e,f*). Ab.E., abaxial epidermis; Sp.M., spongy mesophyll; Pa.M., palisade mesophyll; Ad.E., adaxial epidermis.

**Table 2.** Element concentrations in almond leaves after nanobiofertilization with Fe or B under growth chamber conditions. The data are means ± s.e. (*n* = 5). Significance level: $^{**}p < 0.01$; $^{***}p < 0.001$. For each element, different letters show significant differences according to Duncan's test at $p < 0.05$.

| | | fertilizer type applied | | | |
|---|---|---|---|---|---|
| leaves | deficient plant | control plant | nanofertilizer | free element | *p*-value |
| Fe (µg g$^{-1}$ DW) | 44.8 ± 1.26d | 64.24 ± 5.23c | 256.86 ± 8.29a | 212.70 ± 5.25b | *** |
| B (µg g$^{-1}$ DW) | 23.37 ± 2.65d | 31.27 ± 1.28c | 211.46 ± 4.43a | 112.89 ± 3.92b | *** |
| apex | | | | | |
| Fe (µg g$^{-1}$ DW) | 21.34 ± 1.28c | 79.81 ± 3.28a | 80.11 ± 4.97a | 57.23 ± 5.52b | ** |
| B (µg g$^{-1}$ DW) | 15.26 ± 2.49c | 31.27 ± 2.95b | 121.09 ± 9.98a | 36.21 ± 2.76b | ** |

**Table 3.** Element concentrations in almond leaves after nanobiofertilization with Fe or B under field conditions. The data are means ± s.e. (*n* = 15). Significance level: $^{***}p < 0.001$. For each element, different letters show significant differences according to Duncan's test at $p < 0.05$.

| | | fertilizer type applied | | |
|---|---|---|---|---|
| element | control | nanofertilizer | free element | *p*-value |
| Fe (µg g$^{-1}$ DW) | 67.83 ± 2.34c | 125.70 ± 5.45a | 108.29 ± 4.71b | *** |
| B (µg g$^{-1}$ DW) | 26.62 ± 1.89c | 70.62 ± 3.93a | 34.9 ± 2.34b | *** |

although it was especially intense in the upper part of the leaf, in the spongy mesophyll (figure 3*a,d*). In deficient leaves, Fe did not exhibit an extensive distribution; we only observed some light spots around the main vessel in the section shown here (figure 3*c,f*).

The B distribution in the sections is represented in figure 4*a–c*. When B was applied in its free form, there were numerous blue-coloured areas, mainly in both the adaxial and abaxial epidermis and in the central part of the mesophyll tissue (figure 4*b,e*). After application in its encapsulated form, B showed a homogeneous distribution with wide areas of bright blue colour, mainly in the mesophyll where we found a dense blue colouration (figure 4*a,d*). However, in B-deficient leaves, we only observed homogeneous light spots in all sections analysed (figure 4*c,f*).

**Figure 4.** B mapping of cross-sections of *Prunus dulcis* L. leaves by FESEM-X-EDS. The blue colour represents the B signal. (*a*) 24 h after nanofertilizer application. (*b*) 24 h after free element application. (*c*) Deficient. SEM image of almond leaf section (*d,e,f*). Ab.E., abaxial epidermis; Sp.M., spongy mesophyll; Pa.M., palisade mesophyll; Ad.E., adaxial epidermis.

# 4. Discussion

Fertilization studies with nanobio-materials are still scarce. Most of the previous work in this area has been carried out with polymers such as ethylene vinyl acetate, chitosan, alginate, agar or gelatin applied to soil [29–31]. However, foliar fertilization is a promising area to develop this new technology focused on nanoparticles [32]. Therefore, in our study, we investigated the potential use of Fe and B nanobiofertilizers, using proteoliposomes extracted from *Brassica* plants. Previously, *Brassica* proteoliposomes have been used for nanoencapsulation of bioactive compounds and their delivery in animal cells [33]. Also, the nanoencapsulation of Zn for delivery into *Brassica* leaf protoplasts has been studied [27]. Based on these findings, the use of proteoliposomes encapsulating mineral nutrients was studied in this work.

Regarding the characterization of our system, the entrapment efficiency (EE) reached was 76% for Fe and 80% for B. Optimal encapsulation is a key factor for an efficient delivery system [34], and in a more efficient system, less fertilization will be necessary, which entails economic savings. Our values are of the same order as those obtained in other work. For example, values of EE between 50 and 90% were obtained when encapsulating a pesticide in a polymeric nanoparticle [35], an EE of 72% was reached in our previous work in which Zn was encapsulated in vesicles from broccoli plants [27], and with nanoparticles based on chitosan, EE values of 50–70% were achieved for encapsulated herbicides [36–38].

Furthermore, the Pf was measured after encapsulating B and Fe. This value serves as an index, showing the degree to which the vesicles maintain their integrity and vesicular shape and are not affected by the encapsulated elements. Also, the functionality of the vesicles as membranes could be important in the delivery of the encapsulated elements. The Pf values obtained are within the optimal ranges determined in a previous study for similar vesicles without encapsulated elements [39]. Moreover, the protein–lipid character of our vesicles is key in the system stability [39] and the variety of different components integrated in our vesicles could favour the union of the elements and thus their encapsulation. The B-binding capacity of different membrane proteins has been described; specifically, in *Arabidopsis thaliana*, 26 proteins were identified by boronate affinity chromatography [40]. Although further investigation will be required to check if B or Fe binds to the protein present in our vesicles, the fact that there was no detrimental effect on their integrity as carriers after element encapsulation supports their suitability.

The Z potential value indicates the stability of the vesicles in a suspension system (systems with values between −10 mV and +10 mV will experience rapid agglomeration [41]). Therefore, our system can be considered stable regarding aggregation. Furthermore, the functionality and permeability of the vesicles did not change with element encapsulation.

One of the most important aspects of foliar fertilization is the penetration efficiency. The entry pathways have been a controversial point due to the physico-chemical properties of the compounds applied and the leaf surface [7]. Thus, some studies have indicated that the stomata are a major point

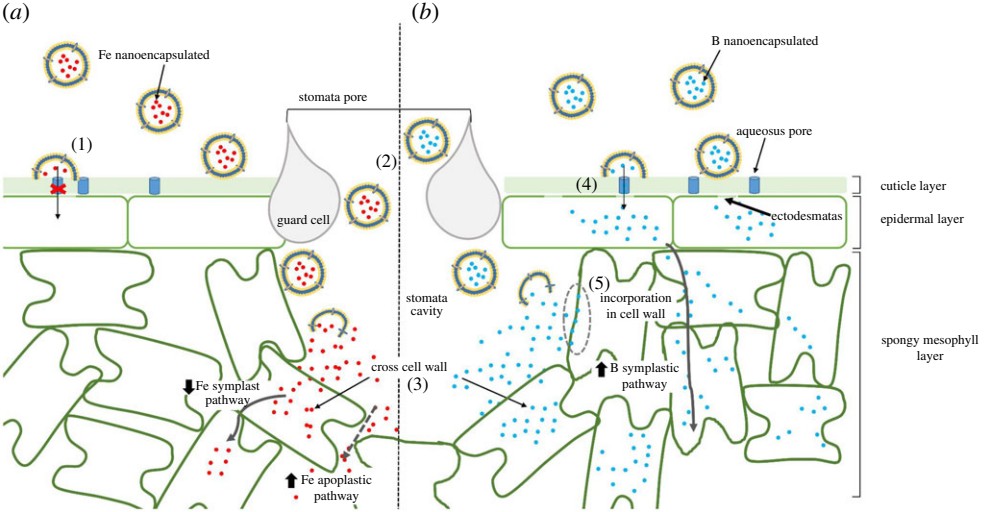

**Figure 5.** Schematic diagram of the longitudinal section of a leaf and the distribution of elements encapsulated in vesicles applied by foliar fertilization. (*a*) The iron (Fe, red circles) encapsulated in vesicles does not cross the cuticle (1), it only passes into the inner layers by entering the stoma (2), where the Fe is released. Fe can across the cell wall (3) and continue by the symplastic pathway (grey continuous arrow) or follow the apoplastic pathway (grey discontinuous arrow). (*b*) Boron (B, blue circles) encapsulated across the cuticle through the aqueous pores (blue cylinders) and the cell wall through ectodesmatas (4), to follow a symplastic pathway (grey continuous arrow). Vesicles with boron pass into the stomata cavity and B is released and incorporated into the cell wall (5) or crosses it (3).

of entry [11,42,43]. However, other studies have suggested cuticular pathways: through pores, when the particles are less than 50 nm in diameter, or via trans-cuticle diffusion or disruption (direct), or both [44]. In this sense, our vesicles are structures with a low external charge, as shown by the Z potential values around −20 mV, and so could avoid electromagnetic interaction with stomatal and cuticular components. Also, the vesicles are around 300 nm in size, with high homogeneity, and so could enter the stomatal pore, which has a width range of 500–1000 nm [12]. For vesicles of this size, it would be difficult to enter by diffusion through the cuticle, a process that has been reported to occur for Zn-coated PVP nanoparticles [44]. Thus, the fact that the concentration of Fe determined in almond leaves was higher after Fe-nanobiofertilizer application, relative to free Fe, indicates higher efficiency of entry through stomata, not a different entry pathway. In this regard, it has been reported that the entry of free $FeSO_4$ applied to peach trees was blocked when ABA was applied to close the stomata [11]. Therefore, when the proteoliposomes reach the stomatal pore they could travel through the apoplast to deliver micronutrients near the cells (figure 5). These elements could then cross the cell wall through selective pores, with a size smaller than 0.1 nm, that allow the entry of ions [45]. This could produce areas with high mineral concentrations, and the TEM images in our experiment show many invaginations in the plasma membrane after treatment application. The leaf cells could enhance the protein concentration selectively in specific parts of the plasma membrane, as has been shown in animal cells as a response to the intake of certain molecules [46]. The invaginations observed here could be used to increase the surface area for contact between elements and specific transporters, to boost their uptake. This interesting point should be investigated.

The Fe distribution images and apex concentrations show a relative mobility of this element among the leaf cell layers. Contrastingly, previous studies indicated that Fe mobility was low and the effect of foliar Fe fertilization was only local [7,47–49]. The leaf cells in Fe-deficient plants are hungry for Fe and once Fe has crossed the cell wall, it is immobilized. Nevertheless, in the present work, the Fe concentration was higher in the upper spongy mesophyll, close to vessels, as was shown also in peach and tomato plants [11,50]. This indicates that intra-tissue Fe movement is symplastic in leaves, and that Fe could be transported to other plant organs. Also, this movement was enhanced by nanofertilizer application, possibly because the vesicles move through the apoplasm, mainly in the space between spongy mesophyll cells.

The leaf B concentrations recorded in our experiment show the vesicles to be twice as efficient as free B. This difference cannot be explained only by charge interactions in the stomatal pore. Then, the results could indicate other pathways of B entry into leaves when vesicles were applied. In this regard, previous

experiments showed that the cuticle is not completely impermeable to solutes [51]. One explication could be that the encapsulated elements crossed through the small pores in the cuticle (called aqueous pores) localized along the entire surface of the leaves, which were described by Schönherr [51], since B can pass through astomatous cuticle membranes [9]. These pores are filled with water and have a diameter of approximately 4–6 nm, while B ions are smaller than 0.1 nm [45]. The fusion of vesicles with the leaf surface could allow B to cross the cuticle through the aqueous pores. However, when free B was applied, the flow through the pore would have been broken due to the immediate development of dryness on the leaf surface (figure 5). The fusion of the vesicles with the lipid outer layer would leave the B in direct contact with the cuticle and aqueous pores, with the lipid coating acting as a protective layer, maintaining the enhanced humidity across both parts of the pore and facilitating the flow of B through it. This protective effect and process could have occurred under both sets of conditions in our experiment (growth chamber and field). This effect was observed recently for Zn nanoparticles in two different types of leaf, wheat and sunflower, for which the authors indicated that coating may facilitate particle adhesion to leaves, providing a long-term source of uptake of Zn ions into leaves [52]. Furthermore, this process probably did not occur with the Fe vesicles since some elements (such as Fe), due to their chemical properties, can compete for water, causing dehydration of the pore and cutting the flow between the two sides of the structure [53].

The mapping shows that once the B had crossed the cuticle it became widely distributed throughout the leaf cell layers and moved to other parts of the plant. Previous studies in some plants, such as members of the genus *Prunus* [19] and kiwi trees [54], indicated that B applied foliarly had high mobility due to the high sorbitol or polyol content of the preparation applied, that could prevent B from quickly adhering to the cell wall. Our study corroborated this since B applied as $H_3BO_3$ had great mobility; however, its transfer to other organs such as the apex was strongly increased when B encapsulated in vesicles was applied. We suggest that the B that crossed the cuticle through the pores quickly passed through the cell wall of the epidermal cells through ectodesmatas, areas (only found in epidermal cells) where the cell wall is less dense and where the exchange is faster [55,56]. Related to this, the invaginations found in the plasma membrane after nanovesicles application could have been provoked by the massive entry of B into the apoplastic area inside the cell wall. So, this B might have been taken up, translocated symplastically across different leaf layers and loaded into the phloem without leakage, as happens in roots [17]. On the other hand, part of the B released from the vesicles would have been retained in the cell walls since, under deficiency, B is quickly retained here [13,54,57]. Another part would bind to sorbitol, as found by Brown & Shelp [19] in *Prunus* spp., enabling it to move to other parts of the leaf, as revealed in our mapping images, as well as to other parts of the plant such as the apex, where the B concentration increased after vesicles or free B application.

# 5. Conclusion

We have shown that nanoencapsulation of micronutrients, in this case B and Fe, into vesicles is a novel nanotechnique that could be applied for highly efficient foliar fertilization. The advantage of our system is that plasma membranes-based vesicles are a compatible material that takes advantage of the natural transport mechanisms of the plant. Therefore, we found an enhanced effect, greater than that of conventional foliar fertilization. This does not happen in the case of fertilization with inorganic carriers, which could even have negative effects on plants [58]. Hence, our results show increased penetrability through stomatal pores for both Fe and B. Also, the results for B suggest that it could enter the leaf through the aqueous pores, producing a greater distribution, both intra-leaf and in other parts of the plant. The results for Fe support the idea that Fe-nanobiofertilization enables the internal movement of this element.

Data accessibility. Our data are deposited at the Dryad Digital Repository: https://doi.org/10.5061/dryad.cz8w9gj13.
Authors' contribution. J.J.R. and M.C. were involved in conception and design; J.J.R. and L.Y.-M. were involved in analysis and interpretation of data and drafting of the article; M.C. was involved in critical revision of the article for important intellectual content; J.J.R., L.Y.-M. and M.C. final approval of the article; J.J.R., L.Y.P. and A.M.-A. were involved in collection, assembly of data and statistical expertize; M.C. was involved in obtaining of funding; M.C. was involved in administrative, technical or logistic support.
Competing interests. The authors declare that they have no known competing financial interests or personal relationships that could have appeared to influence the work reported in this paper.

Funding. This work was funded by the Spanish Ministerio de Ciencia, Innovación y Universidades (AGL2016-80247-C2-1-R and RTC-2017-6544-2) and with a grant for L.Y.-M. (FPU-17/0226).
Acknowledgements. The authors thank Dr. David Walker for the correction of the English.

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
