## [Reviewer comments · Royal Society Open Science]

Review History

RSOS-200905.R0 (Original submission)

Review form: Reviewer 1

Is the manuscript scientifically sound in its present form?

Yes

Are the interpretations and conclusions justified by the results?

Yes

Is the language acceptable?

Yes

Do you have any ethical concerns with this paper?

No

Have you any concerns about statistical analyses in this paper?

No

Recommendation?

Accept with minor revision (please list in comments)

Comments to the Author(s)

Rios and collaborators manuscript studied the potential for foliar NP delivery to meet B and Fe deficiencies in *Prunus* trees. They have studied these impacts on-field and on greenhouse-grown trees, and compared the efficiency of free vs encapsulated ions. The authors compared element repartition on leaves cross-sections, element accumulation in treated and non-treated leaves, and are proposing different mechanisms involved for the differences observed. I think the latter could get more discussion within the manuscript, as described later. Other than that this is a well-written manuscript, with comprehensive results and adequate discussion. I think this manuscript is suitable for publication in Royal Society Open Science. I just have minor revision questions, as described below.

1. Page 4 line 32: What kind of substrate was used?
2. Page 4 line 59: please indicate (approximately) the growth period that needed the trees to develop Fe and B deficiencies symptoms
3. Page 5 Line 17 and line 40: Could the authors indicate the volume of suspension applied on the leaves? Was there a lot of dripping from the leaves? Was the soil covered to avoid contamination of the substrate?
4. Page 9 line 28: DLS does not measure the average diameter, but the average hydrodynamic diameter.
5. Page 9 line 33: please provide the pH of the solution, as the zeta potential is directly influenced by the pH
6. Page 9 line 54: How can the authors be sure that this plasma membrane invagination is not an artifact of preparation? Did the authors repeat these observations on duplicates and observed similar effects?
7. Page 10 line 3: What leaves were used for total element concentration measurements? If the leaves used are the ones that have been directly treated with the suspension, it is not surprising that the concentration increased, and it does not necessarily relate to the element bioavailability, but to its adhesion in surface and/or differential uptake and mobility. These would deserve more discussion and clarification.
8. Page 10 Elements mapping: It is difficult to know if this Fe element distribution is due to the foliar treatment, or if a control leaf would have undergone a similar Fe distribution. Could the authors provide an image of control (deficient and non-deficient) leaves?
9. Page 13 line 3ff: How can the authors be sure that this is not a different pathway of entrance? Some studies hypothesized that hydrophobic compounds could directly cross the cuticle[1]. But others didn't in different systems[2]. These vesicles may be more hydrophobic/more mobile through the cuticle of apple trees? Another hypothesis might be that ions have more affinity to the surface structures of leaves than the vesicles, allowing them to be more mobile. Finally, the authors used a surfactant (btw, what surfactant was used?), that may have altered the cuticle integrity, which could modulate the interactions at the interface that modulate NP or ionic uptake. And the plasma membrane invagination may as well be a result of the surfactant used, as the water control did not contain any. I think an additional discussion is needed here on these aspects, and that ruling out these processes and just stating about stomatal uptake might be a false statement.
10. Page 14 line 30: Could this difference between B and Fe uptake when added to the vesicle be due to different affinities between the proteins and the different ions? See for instance this study that highlights the large differences of logK between different ions and BSA proteins (I know that is not the best model here, but this is just to highlight the process).[3]
11. Did the authors notice an improvement in these nutrient deficiencies symptoms after the different applied treatments?

Cited references

- (1) Avellan, A.; Yun, J.; Zhang, Y.; Spielman-Sun, E.; Unrine, J. M.; Thieme, J.; Li, J.; Lombi, E.; Bland, G.; Lowry, G. V. Nanoparticle Size and Coating Chemistry Control Foliar Uptake Pathways, Translocation, and Leaf-to-Rhizosphere Transport in Wheat. *ACS Nano* 2019, 13 (5), 5291–5305. <https://doi.org/10.1021/acsnano.8b09781>.
- (2) Read, T. L.; Doolette, C. L.; Li, C.; Schjoerring, J. K.; Kopittke, P. M.; Donner, E.; Lombi, E. Optimising the Foliar Uptake of Zinc Oxide Nanoparticles: Do Leaf Surface Properties and

Particle Coating Affect Absorption? *Physiol. Plant.* 2020, ppl.13167.
<https://doi.org/10.1111/ppl.13167>.

(3) Topală, T.; Bodoki, A.; Oprean, L.; Oprean, R. Bovine Serum Albumin Interactions with Metal Complexes. *Clujul Med.* 2014, 87 (4), 5. <https://doi.org/10.15386/cjmed-357>.

Decision letter (RSOS-200905.R0)

Dear Miss Carvajal:

Title: Nanobiofertilization as a novel technology for highly efficient foliar application of Fe and B in almond trees

Manuscript ID: RSOS-200905

Thank you for submitting the above manuscript to Royal Society Open Science. On behalf of the Editors and the Royal Society of Chemistry, I am pleased to inform you that your manuscript will be accepted for publication in Royal Society Open Science subject to minor revision in accordance with the referee suggestions. Please find the reviewers' comments at the end of this email.

The reviewers and handling editors have recommended publication, but also suggest some minor revisions to your manuscript. Therefore, I invite you to respond to the comments and revise your manuscript.

Because the schedule for publication is very tight, it is a condition of publication that you submit the revised version of your manuscript before 11-Sep-2020. Please note that the revision deadline will expire at 00.00am on this date. If you do not think you will be able to meet this date please let me know immediately.

- 1) A text file of the manuscript (tex, txt, rtf, docx or doc), references, tables (including captions) and figure captions. Do not upload a PDF as your "Main Document".
- 2) A separate electronic file of each figure (EPS or print-quality PDF preferred (either format should be produced directly from original creation package), or original software format)
- 3) Included a 100 word media summary of your paper when requested at submission. Please ensure you have entered correct contact details (email, institution and telephone) in your user account

4) Included the raw data to support the claims made in your paper. You can either include your data as electronic supplementary material or upload to a repository and include the relevant doi within your manuscript

5) All supplementary materials accompanying an accepted article will be treated as in their final form. Note that the Royal Society will neither edit nor typeset supplementary material and it will be hosted as provided. Please ensure that the supplementary material includes the paper details where possible (authors, article title, journal name).

Kind regards,
Dr Ellis Wilde
Publishing Editor, Journals

On behalf of the Subject Editor Professor Anthony Stace and the Associate Editor Dr Nadia Martinez Villegas.

RSC Subject Editor

Comments to the Author:

The research presented in this draft is original and of interest to RSOS audience, however additional details and clarification are needed in order to improve the Material and Methods section and fully achieve a scientifically sound experimental design capable of ruling out any possible artifacts of preparation. For doing so, please see comments from the reviewer.

Additionally, to convincingly defend the hypotheses for the pathways of entrance of Fe and B, a more comprehensive discussion is needed in the manuscript.

RSC Associate Editor

Comments to the Author:

(There are no comments.)

Reviewer comments to Author:

Reviewer: 1

Comments to the Author(s)

Rios and collaborators manuscript studied the potential for foliar NP delivery to meet B and Fe deficiencies in Prunus trees. They have studied these impacts on-field and on greenhouse-grown

trees, and compared the efficiency of free vs encapsulated ions. The authors compared element repartition on leaves cross-sections, element accumulation in treated and non-treated leaves, and are proposing different mechanisms involved for the differences observed. I think the latter could get more discussion within the manuscript, as described later. Other than that this is a well-written manuscript, with comprehensive results and adequate discussion. I think this manuscript is suitable for publication in Royal Society Open Science. I just have minor revision questions, as described below.

1. Page 4 line 32: What kind of substrate was used?
2. Page 4 line 59: please indicate (approximately) the growth period that needed the trees to develop Fe and B deficiencies symptoms
3. Page 5 Line 17 and line 40: Could the authors indicate the volume of suspension applied on the leaves? Was there a lot of dripping from the leaves? Was the soil covered to avoid contamination of the substrate?
4. Page 9 line 28: DLS does not measure the average diameter, but the average hydrodynamic diameter.
5. Page 9 line 33: please provide the pH of the solution, as the zeta potential is directly influenced by the pH
6. Page 9 line 54: How can the authors be sure that this plasma membrane invagination is not an artifact of preparation? Did the authors repeat these observations on duplicates and observed similar effects?
7. Page 10 line 3: What leaves were used for total element concentration measurements? If the leaves used are the ones that have been directly treated with the suspension, it is not surprising that the concentration increased, and it does not necessarily relate to the element bioavailability, but to its adhesion in surface and/or differential uptake and mobility. These would deserve more discussion and clarification.
8. Page 10 Elements mapping: It is difficult to know if this Fe element distribution is due to the foliar treatment, or if a control leaf would have undergone a similar Fe distribution. Could the authors provide an image of control (deficient and non-deficient) leaves?
9. Page 13 line 3ff: How can the authors be sure that this is not a different pathway of entrance? Some studies hypothesized that hydrophobic compounds could directly cross the cuticle[1]. But others didn't in different systems[2]. These vesicles may be more hydrophobic/more mobile through the cuticle of apple trees? Another hypothesis might be that ions have more affinity to the surface structures of leaves than the vesicles, allowing them to be more mobile. Finally, the authors used a surfactant (btw, what surfactant was used?), that may have altered the cuticle integrity, which could modulate the interactions at the interface that modulate NP or ionic uptake. And the plasma membrane invagination may as well be a result of the surfactant used, as the water control did not contain any. I think an additional discussion is needed here on these aspects, and that ruling out these processes and just stating about stomatal uptake might be a false statement.
10. Page 14 line 30: Could this difference between B and Fe uptake when added to the vesicle be due to different affinities between the proteins and the different ions? See for instance this study that highlights the large differences of logK between different ions and BSA proteins (I know that is not the best model here, but this is just to highlight the process).[3]
11. Did the authors notice an improvement in these nutrient deficiencies symptoms after the different applied treatments?

Cited references

- (1) Avellan, A.; Yun, J.; Zhang, Y.; Spielman-Sun, E.; Unrine, J. M.; Thieme, J.; Li, J.; Lombi, E.; Bland, G.; Lowry, G. V. Nanoparticle Size and Coating Chemistry Control Foliar Uptake Pathways, Translocation, and Leaf-to-Rhizosphere Transport in Wheat. *ACS Nano* 2019, 13 (5), 5291-5305. <https://doi.org/10.1021/acsnano.8b09781>.
- (2) Read, T. L.; Doolette, C. L.; Li, C.; Schjoerring, J. K.; Kopittke, P. M.; Donner, E.; Lombi, E. Optimising the Foliar Uptake of Zinc Oxide Nanoparticles: Do Leaf Surface Properties and Particle Coating Affect Absorption? *Physiol. Plant.* 2020, ppl.13167. <https://doi.org/10.1111/ppl.13167>.

(3) Topală, T.; Bodoki, A.; Oprean, L.; Oprean, R. Bovine Serum Albumin Interactions with Metal Complexes. Clujul Med. 2014, 87 (4), 5. <https://doi.org/10.15386/cjmed-357>.

Author's Response to Decision Letter for (RSOS-200905.R0)

See Appendix A.

RSOS-200905.R1 (Revision)

Review form: Reviewer 1

Is the manuscript scientifically sound in its present form?

Yes

Are the interpretations and conclusions justified by the results?

Yes

Is the language acceptable?

Yes

Do you have any ethical concerns with this paper?

No

Have you any concerns about statistical analyses in this paper?

No

Recommendation?

Accept as is

Comments to the Author(s)

I think the authors addressed all of my comments and would recommend the manuscript for publication as it is. The authors and/or the journal should make a final check for English grammar mistakes and spelling errors, as a lot of them seem to remain.

Decision letter (RSOS-200905.R1)

Dear Miss Carvajal:

Title: Nanobiofertilization as a novel technology for highly efficient foliar application of Fe and B in almond trees

Manuscript ID: RSOS-200905.R1

It is a pleasure to accept your manuscript in its current form for publication in Royal Society Open Science. The chemistry content of Royal Society Open Science is published in collaboration with the Royal Society of Chemistry.

On behalf of the Subject Editor Professor Anthony Stace and the Associate Editor Dr Nadia Martinez Villegas.

RSC Associate Editor:
Comments to the Author:
(There are no comments.)

RSC Subject Editor:
Comments to the Author:
(There are no comments.)

Reviewer(s)' Comments to Author:
Reviewer: 1

Comments to the Author(s)

I think the authors addressed all of my comments and would recommend the manuscript for publication as it is. The authors and/or the journal should make a final check for English grammar mistakes and spelling errors, as a lot of them seem to remain.

Appendix A

Comments from referee

1. *Page 4 line 32: What kind of substrate was used?*

The composition of the substrate used has been added.

2. *Page 4 line 59: please indicate (approximately) the growth period that needed the trees to develop Fe and B deficiencies symptoms*

we have added the days need until deficiencies symptoms appeared (20 days).

3. *Page 5 Line 17 and line 40: Could the authors indicate the volume of suspension applied on the leaves? Was there a lot of dripping from the leaves? Was the soil covered to avoid contamination of the substrate?*

We have added the amount applied per leaf. As it was small amount, it did not produce any drip.

In the other hand, the hydroponic system, which was used in the experiments, has a plastic covers that hold the plants. That cover avoided the contamination of the solution in the event that a drop was fallen.

4. *Page 9 line 28: DLS does not measure the average diameter, but the average hydrodynamic diameter.*

Right. We have added 'hydrodynamic'

5. *Page 9 line 33: please provide the pH of the solution, as the zeta potential is directly influenced by the pH.*

The pH was 6.5. It has been included

6. *Page 9 line 54: How can the authors be sure that this plasma membrane invagination is not an artefact of preparation? Did the authors repeat these observations on duplicates and observed similar effects?*

We are sure of our result due to the images were repeated in the all cross sections of the different blocks from various leaves in 2 different experiments. It has been included in the section 2.8 Transmission electron microscopy (TEM) in Material and Methods

7. *Page 10 line 3: What leaves were used for total element concentration measurements? If the leaves used are the ones that have been directly treated with the suspension, it is not surprising that the concentration increased, and it does not necessarily relate to the element bioavailability,*

but to its adhesion in surface and/or differential uptake and mobility. These would deserve more discussion and clarification.

Yes, it is correct the treatments were applied on same leaves that used to measure element concentration. However, as the leaves were thoroughly washed to avoid the remains adhering to the surface, the element analysis must be due to the uptake and mobility. More discussion has been included.

8. Page 10 Elements mapping: It is difficult to know if this Fe element distribution is due to the foliar treatment, or if a control leaf would have undergone a similar Fe distribution. Could the authors provide an image of control (deficient and non-deficient) leaves?

We have included the pictures of deficient Fe and B in the figures 3 and 4, for comparison. It can be observed that deficient leaves showed less spots. Also, some sentences have been included in results to describe these new images.

9. Page 13 line 3ff: How can the authors be sure that this is not a different pathway of entrance? Some studies hypothesized that hydrophobic compounds could directly cross the cuticle [1]. But others didn't in different systems [2]. These vesicles may be more hydrophobic/more mobile through the cuticle of apple trees? Another hypothesis might be that ions have more affinity to the surface structures of leaves that the vesicles, allowing them to be more mobile.

The fact that the vesicles could more easily cross the cuticle due to their hydrophobicity could be rejected due to the size of the vesicles. However, we think that they could adhere to the cuticle more easily. That is why we suggest different pathways for the 2 different elements depending on their size and charge. Fe cannot use this cuticle pore as discussed.

The application of the elements in free form has lower absorption than in encapsulated form. Therefore, from the results of the experiments we suggest that the free elements can be retained to the leaf surface, preventing its rapid absorption and decreasing the efficiency of absorption. In the experiments the concentrations applied for each element were the same, in free or encapsulated form, so we can check the efficiency rate in their absorption.

Also, we have made some changes in discussion to improve it with references provided by the reviewer.

Finally, the authors used a surfactant (btw, what surfactant was used?), that may have altered the cuticle integrity, which could modulate the interactions at the interface that modulate NP or ionic uptake. And the plasma membrane invagination may as well be a result of the surfactant used, as the water control did not contain any. I think an additional discussion is needed here on these aspects, and that ruling out these

processes and just stating about stomatal uptake might be a false statement.

We have added the type of surfactant in material and methods.

As the control treatment also contained surfactant, the invaginations are due to nanoencapsulated samples. Clarification has been included in material and methods.

The conclusion according to the Fe encapsulated entrance is due to our previous results in which

We have made some changes in discussion to improve it. In the case of Fe, the entry has been demonstrated by previous studies such as those of Rios et al. 2016, which is the stoma the way of entry. Comparing with the result obtained in the paper suggested by the referee, the nanoparticles were smaller (50 nm) that could allow cuticle entrance.

10. Page 14 line 30: Could this difference between B and Fe uptake when added to the vesicle be due to different affinities between the proteins and the different ions? See for instance this study that highlights the large differences of logK between different ions and BSA proteins (I know that is not the best model here, but this is just to highlight the process).

It is a good observation. We have included a sentence in the discussion. Although we did not find, a detrimental effect on their integrity as carriers after element encapsulation, we cannot discard a different affinity between proteins and Fe or B. This aspect need to be investigated in the future.

11. Did the authors notice an improvement in these nutrient deficiencies symptoms after the different applied treatments?

We could observed an improve in leaves colour to iron deficiency, but as the result was a short term, due to the samples were collected 24h after application, the improvement of the symptoms was very light. But, these experiments are going to be carried out.